# Dog Population & Dog Sheltering Trends in the United States of America

**DOI:** 10.3390/ani8050068

**Published:** 2018-04-28

**Authors:** Andrew Rowan, Tamara Kartal

**Affiliations:** 1Chief Scientific Officer, The Humane Society of the United States, 1255 23rd Street, NW, Washington, DC 20037, USA; 2Companion Animal Division, Humane Society International, 1255 23rd Street, NW, Washington, DC 20037, USA; tkartal@hsi.org

**Keywords:** humane dog management, shelter statistics, sterilization, human-canine relationship

## Abstract

**Simple Summary:**

The pet overpopulation problem in the United States has changed significantly since the 1970s. The purpose of this review is to document these changes and propose factors that have been and are currently driving the dog population dynamics in the US. In the 1960s, about one quarter of the dog population was still roaming the streets (whether owned or not) and 10 to 20-fold more dogs were euthanized in shelters compared to the present. We present data from across the United States which support the idea that, along with increased responsible pet ownership behaviors, sterilization efforts in shelters and private veterinary hospitals have played a role driving and sustaining the decline in unwanted animals entering shelters (and being euthanized). Additionally, data shows that adoption numbers are rising slowly across the US and have become an additional driver of declining euthanasia numbers in the last decade. We conclude that the cultural shift in how society and pet owners relate to dogs has produced positive shelter trends beyond the decline in intake. The increased level of control and care dog owners provide to their dogs, as well as the increasing perception of dogs as family members, are all indicators of the changing human-dog relationship in the US.

**Abstract:**

Dog management in the United States has evolved considerably over the last 40 years. This review analyzes available data from the last 30 to 40 years to identify national and local trends. In 1973, The Humane Society of the US (The HSUS) estimated that about 13.5 million animals (64 dogs and cats per 1000 people) were euthanized in the US (about 20% of the pet population) and about 25% of the dog population was still roaming the streets. Intake and euthanasia numbers (national and state level) declined rapidly in the 1970s due to a number of factors, including the implementation of shelter sterilization policies, changes in sterilization practices by private veterinarians and the passage of local ordinances implementing differential licensing fees for intact and sterilized pets. By the mid-1980s, shelter intake had declined by about 50% (The HSUS estimated 7.6–10 million animals euthanized in 1985). Data collected by PetPoint over the past eight years indicate that adoptions increased in the last decade and may have become an additional driver affecting recent euthanasia declines across the US. We suspect that sterilizations, now part of the standard veterinary care, and the level of control of pet dogs exercised by pet owners (roaming dogs are now mostly absent in many US communities) played an important part in the cultural shift in the US, in which a larger proportion of families now regard their pet dogs as “family members”.

## 1. Introduction

There has been a lot of public commentary on the evolution of dog management in the United States but very few analyses of the vast amount of relevant data, most of it from “grey” sources, in the United States. The data available are not considered particularly robust, consisting as they do of reports by individual shelters and analyses in the “grey” literature (e.g., Shelter Sense Magazine from The Humane Society of the US (The HSUS), Animal People newsmagazine, etc.), but these data offer important insights into national sheltering trends. A recent review of the literature on companion animal population demographics (a total of 931 reviews and research reports) finds that the frequency of relevant scientific publications on animal sheltering has increased from 5–10 a year in the 1970s to 50 or more a year in the last decade [1]. However, only a few of these have reported on national or regional shelter trends in the USA.

We aim to remedy this omission in this review. We will incorporate data from various sources to construct an analysis of trends in shelter demographics and in US dog populations starting in the 1970s.

### Data Sourcing

As indicated, reports on US shelter numbers as well as shelter demographic studies in the literature are few and far between, although the number of papers have increased in the last 30 years (see Kay et al. 2017 [1]). One of the problems with the available data on shelter intake and outcomes is that it has never` been collected and reported consistently from 1970 to the present. Even data on the number of companion animals in the US is subject to methodological problems. The two main surveys by the American Veterinary Medical Association (AVMA—surveyed every five years since 1986) and the American Pet Products Association (APPA—surveyed every two years since 1988) differ significantly in the estimated national populations of companion dogs and cats. There have been several efforts over the past fifty years to collect national shelter data (the most recent is Shelter Animals Count—https://shelteranimalscount.org/) but none have succeeded in either producing an accurate number of shelters (physical structures that house animals) in the country or the national intake and outcomes of dogs and cats into these shelters.

However, we argue that it is possible to piece together data from various sources (see Appendix A for more on this issue) including:(a)individual shelters;(b)previous attempts to track national shelter trends;(c)state reports of shelter numbers; other attempts to track shelter trends;(d)from the commercial software vendor, PetHealth.

PetHealth’s shelter software app, PetPoint, is a free shelter app that also collects and stores data from participating shelters (organizations that have buildings that house shelter animals) and animal rescues. PetHealth publishes compiled monthly reports of intake and outcomes from approximately half of the entities using the software. Their monthly reports provide what we argue is a reasonably accurate and internally consistent picture of sheltering trends in the USA from 2009 to the present.

However, there are issues with the Petpoint data. Only relatively few municipal and city shelters (out of an estimated 1400 in the USA—estimates from unpublished surveys conducted by The HSUS) use the Petpoint app because the data on people adopting dogs from the shelter are stored on PetHealth servers and public officials are loath to outsource data collection and storage. Furthermore, the Petpoint data reflect a compilation from 900 to 1300 entities (the number is steadily increasing) although Petpoint has over 2000 users. The reported monthly data are compared to the same month a year earlier so a shelter or rescue has to be using the app for both months in order to be included in the composite tally. We have “standardized” the monthly data to “represent” 1000 shelters and rescues (when the number of shelters and rescues for the month is lower than 1000, then the standardized number will be increased and when the composite number reflects more than 1000 entities, the standardized number will be lower). We do not know if this “standardization” is valid because we do not know how the entities covered change from month to month. Finally, we assume that the “standardized” data represents approximately 20% of total shelter and rescue intake and outcome. This assumption is based on the trends and national estimates from other sources and is not easily validated. We intend to keep working on the current data sets and new ones that are being produced to produce better national estimates of absolute shelter and rescue intake and outcomes.

While the various data sources vary in their reliability and comprehensiveness, we maintain that it is possible to use them to construct a picture of national trends from 1970 to the present. It is not possible to determine the causes for those trends from the available data, but we suggest some possible reasons for the huge decline in shelter animal intake in the USA since 1970.

We acknowledge that the data in this review are not standardized across all locations and sources. However, the trends appear to be more or less the same on all scales, from individual shelters and from state compilations. We argue that it is not the number from one year to the next, but rather the longer term trends within one scale or location (e.g., a single shelter or an individual state) and between different scales or locations (e.g., compare trends between individual shelters and states), that provides a reasonably accurate picture of national developments. Further, we believe that all data sources in this review contain their own biases and systemic errors but they produce similar trends over time. As such, we believe the data demonstrate the overall national shelter trend from 1970 onwards in the USA.

## 2. National Trends in Animal Shelter Demographics from 1970 to 2010

In the USA, a network of animal shelters was established in the late 19th and early 20th centuries. In 1910, McCrea [2] reported there were around 500 humane societies and SPCAs in the USA in September 1908 (humane societies worked on both animal and child protection, SPCAs worked only on animal protection). In 1959, Robert Chenoweth [3], then the President of The HSUS, estimated there were only 350 or so active shelters in the USA. This number had grown to around 3500 shelters in 3100 counties by 2015 (based on unpublished surveys by The HSUS).

In a 1973 survey of shelters, The HSUS estimated that 13.5 million dogs and cats were euthanized nationwide by shelters. This worked out to around 64 dogs and cats per 1000 people. This total was equal to around 20% of the owned dog (about 35 million) and cat (around 30 million) populations at the time [4]. In most shelters and pounds, over 90% of the incoming animals were euthanized and the costs of taking in animals, caring for them for three to seven days and then euthanizing them consumed the budgets of animal control agencies and humane societies. Very little money was available to spend on pet sterilization or other preventive programs [5]. Animal control was still traditionally a low budget priority for municipal governments in terms of staffing and enforcement [4] and roaming pets and strays were a serious concern.

In the early 1970’s this problem was highlighted in a flurry of articles, particularly an editorial in Science [6] and a paper in the Bulletin of Atomic Scientists in 1974 by Carl Djerassi [7] (the founder of the birth control pill) and his colleagues. The pet “overpopulation” crisis became a national issue and two meetings by the key stakeholders (including the major national humane and veterinary organizations) in 1974 and 1976 started discussions on more humane and sustainable solutions to the pet overpopulation issue. These meetings gave rise to the legislation, education, and sterilization (LES) project conceived by Phyllis Wright of The HSUS [8].

All three of the major national groups addressing animal sheltering at the time (The HSUS, the American Humane Association, and the National Animal Control Association) promoted local ordinances that enforced responsible pet ownership. Basic requirements included the licensing of dogs and cats with tags attached to their collars and, increasingly, a differentiated licensing fee for intact and sterilized pets. Additionally owners were encouraged to control their animals at all times, breeders were to be licensed and subject to regulations, and there were campaigns to ensure the sterilization of all animals adopted from public and private shelters [8].

### The Success of Sterilization and Differentiated Dog Licensing in the United States of America

Included in general trends in increased responsible pet ownership behaviors were increased sterilization rates for pets, likely a major contributor to the huge decline in shelter euthanasia in the United States. This was facilitated by the establishment of “low-cost” sterilization clinics for pets by municipal animal control and private shelters. Although these clinics performed only a small proportion of the sterilizations of owned dogs and cats, we speculate they also led to an increase in sterilizations at private veterinary clinics, and the establishment of “high-volume, low-cost” specialty veterinary spay/neuter clinics. One of the first of these started in Los Angeles by Dr Mackie in 1976 after the Department of Animal Regulation Services (now Los Angeles Animal Services) set up a [9] municipal clinic in 1971 [4] (see Figure 1 and Figure 2). According to dog licensing data [10], only 10.9% of the licensed dogs in the City of LA were sterilized in 1971. However, in just a few years, this percentage had jumped to 50% (it is now virtually 100%). From an examination of the number of dogs that would have had to be sterilized in Los Angeles, and the number of sterilizations actually performed in the municipal clinic, it appears that 80% or more of the sterilizations from 1970 to 1980 were performed by private veterinary practices [11,12] rather than by the municipal clinic (which averaged around 10,000 sterilizations a year from 1975 onwards).

While there were no controlled studies of the impact of low-cost pet sterilizations by municipal, shelter and private veterinary clinics on animal intake and euthanasia, there were big declines in animal intake (and euthanasia) in many communities during the 1970s. In fact, the typical trend in animal intake and euthanasia featured a big decline (25–40%) in the 1970s, a levelling off in the 1980s, followed by a slower but steady decline from 1990 onwards [18]. Despite the expansion of low cost and high-volume sterilization clinics, Marsh (2010) [5] reports that the overwhelming majority of spay/neuter surgeries in the United States are performed at private veterinary hospitals. While targeted subsidy programs are an essential component of an effective community dog population control plan, private veterinary clinics sterilize an estimated five cats and dogs for every one sterilized through a shelter or subsidy program [5]. In 2005, an estimated 11,000,000 pet sterilizations were performed by private veterinary hospitals, while 2,112,000 were performed through shelters, spay/neuter programs, and feral cat sterilization programs. The high proportion of veterinary clients neutering pets reflects successful efforts by municipal authorities, veterinarians and animal welfare shelters in persuading owners to have their pets sterilized [5].

From 1980 to 1985, The HSUS found that the number of dogs and cats handled by shelters declined by an average of 12% in shelters in communities that imposed differential licensing fees in which owners of unsterilized pets pay a higher fee to license their pet [4]. In a 1982 survey of shelters (a follow-up from the 1973 survey), The HSUS estimated that the dog and cat euthanasia figure had fallen to 7.6–10 million, and many shelters reported declines in animals handled in the 1980′s despite considerable growth in the human populations in their communities. Rowan and Williams (1987) [4] concluded that the pet “overpopulation” problem (i.e., the euthanasia of animals in shelters) decreased from 20% of the total pet population to around 10% of the owned dog and cat population being euthanized nationally in the decade of the 1980s. Figure 3 and Figure 4 below illustrate these trends in dog and cat shelter intakes and subsequent euthanasia rates.

While the reasons for the declines in the 1970s are unclear and perforce speculative, it appears that legislation, education, and sterilization (LES) programs had some impact [4]). Various reports have concluded that the drive to sterilize pet dogs and cats and, more recently, stray cats has been a major factor in the decline in shelter euthanasia (e.g., [18,19,20]). There were further declines in intake and euthanasia after the 1970s and 1980s. The chart below (Figure 3), showing data from Peninsula Humane Society in San Mateo County, California (from 1970 to 1994) and from California state records of shelter intake for the whole of San Mateo county from 1997 to 2015, illustrates some of the changes that were occurring in the 1990s and the 2000s. After levelling off for a few years, dog intake began to decline again in the late 1980s. The number of high-volume sterilization clinics and initiatives has been increasing across the country to address pockets of poverty where dog and cat sterilization rates remain low (e.g., see [5,18,21,22,23,24]).

Overall euthanasia of dogs and cats in US shelters has undergone a steady and rather dramatic decline (Figure 4). However, the earlier data points are not particularly robust. The 1973 and 1982 points were based on surveys of shelter operations around the United States by staff of The HSUS. The 1990 and 2000 datapoints are based on analyses and estimates by a small group of individuals, including one of the authors of this article (ANR), Merritt Clifton and Phil Arkow. The final set of datapoints (2009 to the present) are based on PetPoint reports of monthly intake and outcome totals for around one thousand or more shelters and rescue operations.

Marsh (2010) [5] also suggests that differential licensing (different fees for sterilized and intact pets) has contributed to reducing U.S. shelter animal intakes in recent times. Between 1993 and 2006, after a $45 surcharge was imposed on licenses for intact pets in King County, Washington, the number of cats and dogs admitted to King County Animal Services shelters dropped by 14.6% despite a 21.1% increase in the county’s population during this period [5]. More than 80% of cities and counties in the United States now impose a differential license surcharge [5]. Other programs to combat pet overpopulation include designated spay-neuter practices in stationary and mobile clinics, field operations, shelter services, voucher systems, in-clinic programs provided through private practitioners, and partnerships with veterinary colleges [28]. A hallmark of these programs is the provision of high quality surgeries to large numbers of patients on a regular basis [28]. In some states, such as New Jersey and Massachusetts, the differential dog licensing schemes and low cost spay and neuter surgeries are coordinated and encouraged by state authorities or by state-wide partnership between the veterinary association and animal shelters.

## 3. Recent National Trends (Post 2010)

Adoptions became a factor driving additional decreases in national shelter euthanasia starting around or just before 2010. Prior to this, shelter euthanasia numbers tracked the intake of dogs and cats quite closely [5]. But, from 2010 onwards, it appears that increased adoptions also started to have an effect on euthanasia rates. If one compares the number of dogs adopted and euthanized between 2009 and 2017, as a proportion of dog intake numbers (Figure 5), adoption (rising) and euthanasia (declining) trends have visibly separated. Regression analysis shows that there is a statistically significant negative relationship between the proportion adopted and euthanized (R^2^ = 0.64, *p* < 0.0001) although the actual increase in adoptions is only about 25% of the decline in euthanasia over the same period. We would also note that the Ad Council has been running a national campaign promoting pet adoption since 2009 (https://www.adcouncil.org/Our-Campaigns/Family-Community/Shelter-Pet-Adoption), and together with advertising of pets for adoption by shelters and rescue groups, this campaign may also be influencing shelter adoption numbers since 2009. There is also a negative relationship between the proportion of dogs returned to their owners and euthanasia (R^2^ = 0.45, *p* < 0.0001) however not as strongly correlated but still statistically significant (see Appendix A for details). Microchips for pet identification became available for use in USA in the mid-1980s but the microchip market in the US has suffered because some companies use ISO compliant chips and some do not. There is now a “universal” scanner that can detect any of the microchips in use but competition between competing standards held back the uptake and use of microchips in the USA [29]. (see August 2009 article, E. Lau: http://news.vin.com/vinnews.aspx?articleId=13737).

We have assumed that the PetPoint data (standardized to be representative of 1000 entities over the period from 2009 to 2017) cover about 20% of all shelter and rescue operation intake and outcome in the United States. Therefore, the total number of dogs handled across the USA is around 5-fold the “standardized” PetPoint numbers. (Note: all the estimates of national shelter intake and outcome are subject to significant uncertainty. However, a recent survey by Woodruff and Smith (see below) produces estimates that are in the range of those produced by our suggestion that national numbers are roughly the standardized PetPoint estimates times a factor of five.) Other characteristics, such as seasonal variations, are also captured in the PetPoint data (see Appendix A).

Woodruff and Smith (2016) [30] generated similar estimates (to these estimates from the Petpoint dataset—Table 1) of national shelter dog intake and outcome data for 2016. Their estimates are based on phone surveys of 2862 animal shelters in 49 states (producing a response rate of 14.4%—or 413 shelters). Woodruff and Smith estimated that 5,532,904 (95% CI = 5,003,528–6,169,579) dogs entered US shelters in 2016. (In order to be identified as a shelter the entity: (1) must accept dogs; (2) must adopt dogs to the public and (3) must house animals in a shelter building). Based on the shelter data submitted to Pet Point in 2016 we estimate that about 4,171,017 dogs entered shelters or animal rescues in 2016. This estimate is 25% lower than that given by Woodruff and Smith (2016) but given the very different assumptions involved in arriving at these two totals, we consider their numbers to be in adequate agreement with our estimate. The major disagreement between the two approaches (Table 1) concerns the number of shelters in the US. Further analysis of this difference is provided in the Appendix A.

But regardless of this discrepancy, we believe that Woodruff and Smith‘s independent estimates support our claim that the PetPoint dataset provides a good source to track the overall shelter trends in the United States from 2009 onwards.

Many approaches have been taken to establish data collection systems and bring together shelters nationwide (see https://www.aspca.org/about-us/aspca-policy-and-position-statements/position-statement-data-collection-reporting) but none produced a widely agreed estimate of national numbers for shelter intake and outcomes. A new and promising platform is Shelter Animals Count. The first annual report was published in 2016 and produced numbers for 2255 participating shelters and rescues, accounting for 1,422,671 dogs and 1,259,381 cats (total cats and dogs: 2,682,052) needing new homes or looking for their pet owners [31]. We hope that this data base will entice enough actual shelters (who handle 80+% of all animal intake) to sign up to produce valuable information and datasets on shelter statistics in the coming years. A preliminary examination of the Florida organizations who have signed up indicates that, as of the end of 2016, only about 30% of the state’s shelters have joined (the remainder are rescue organizations).

## 4. Regional and State Trends in Shelter Demographics

Dog demographics and rates of ownership vary significantly between states in the United States. Statewide shelter numbers and trends are difficult to obtain and mostly unavailable, hence shelter and euthanasia trends are only available for a few states (Table 2). However, relatively reliable dog population numbers are generated by the AVMA and are available for 48 states. (The AVMA has traditionally sent out around 80,000 questionnaires to a sample of households drawn from a panel of 400,000 who have been compiled by a commercial company and who have agreed to participate in surveys.) While New Hampshire and New Jersey have low intake and euthanasia rates and have significantly lower numbers of stray dogs, other states, especially in the South, report much higher numbers. Clifton (2014) [26] has continued to gather shelter data annually over decades and adds some numbers to the perception of a Northeast region with very few adoptable dogs in shelters and an ongoing dog overpopulated South (Table 2). The states in Clifton’s discussion are not randomly selected. He employs a convenience sample of all the states with available shelter data. However, his dataset includes states and shelters representing almost one third of the human population in the United States (totaling 308.7 million people in 2010) and provides an overview of the different situations across the country. New Hampshire euthanized 0.26 dogs per 1000 people in 2012 whereas North Carolina euthanized almost 25 times more dogs per 1000 people (6.45 dogs per 100 humans) in 2013. Nevada euthanized 21 times more dogs (5.39 dogs per 100 humans) and even California euthanized 18 times more dogs (4.69 dogs per 100 humans) than New Hampshire (Table 2 [18]). The Northeast has relatively low dog ownership rates compared to other regions of the country (Table 2).

### 4.1. State Trends

#### 4.1.1. New Jersey

In 1984, the state of New Jersey was the first in the country to address its pet overpopulation problem with a statewide low-income, low-cost, spay/neuter program [34] and continues to provide these services. The number of dogs impounded dropped by 75% from 1984 to 2014, while the number of dogs euthanized has dropped by over 90% over the same period (Figure 6 and Figure 7) (numbers provided by New Jersey Department of Health, Infectious and Zoonotic Diseases, various). Absolute adoption numbers have stayed relatively stable over the past three decades but the percentage of intake adopted rose from 20% in 1984 to about 35% in 1991, stayed relatively stable at 35% from 1991 to 2005 and then started rising and had reached 52% by 2016. Marsh [5] suggests that intake is the main driver in a declining euthanasia trend, however, if we look at shelter trends as proportions of intake it becomes clear that adoptions have started to become another main driver in New Jersey in the last decade (Figure 6). Figure 7 provides a chart of absolute intake and outcome numbers which indicate that adoptions have not increased in absolute terms but as intake numbers decline, the % of dogs adopted will increase.

#### 4.1.2. California

An analysis of dog shelter intake and euthanasia in California from 1997 through 2013 indicates great variation in shelter trends from county to county [13]. Some counties achieve low rates of shelter euthanasia while others are closer to the national average or even higher (Figure 8 and Figure 9). The two charts below show the rate of dog intake and euthanasia per 1000 people for Fresno and San Diego respectively. Both shelters have experienced a similar slope in decrease of intake and euthanasia per 1000 people over the years. However, Fresno’s euthanasia rates are still high at around 12.8 (per 1000 people) compared to San Diego at around 1.5 dogs per 1000 people. In fact, all the coastal counties in California tend to have low euthanasia rates (San Francisco and San Luis Obispo are the lowest at under 2 per 1000) while rates in the inland counties are much higher. It is not clear why this difference exists although the coastal counties tend to be wealthier than the inland counties.

Although the quality of the collected data for counties in California varies, we believe that there are enough years of data to discern trends in animal intake and euthanasia (Figure 10) (Note: the data in Figure 10 is a composite of values developed from trend lines calculated individually for all 58 counties in California from reports on shelter intake and outcomes produced by the California Department of Health and Human Services (see [15], annually since 1997)).

As in other states and across the country, there has been a significant decline in dogs entering shelters (per 1000 people) and in the euthanasia rate (Figure 10).

#### 4.1.3. Michigan

Since 2000, every Michigan shelter has to be licensed and has to report their shelter data annually under the Pet Shop, Dog Pounds, and Animal Shelters Act, 1996. Based on the annual reports published by the Michigan Department of Agriculture and Rural Development, an average of 85% of Michigan shelters have reported their statistics in the last years. Figure 11 illustrates the downward trend in intake and euthanasia. In 2003, Bartlett et al. [35] used the same Michigan State law to obtain data from all Michigan shelters. They found that 5.65 dogs per 1000 people were euthanized in 2003 similar to the 5.69 dogs per 1000 people in 2005 (Figure 11). Since 2006 the euthanasia rate has been falling and reached 2.3 dogs per 1000 in 2013 [36].

#### 4.1.4. Ohio

Lord et al., (2006) [37] at Ohio State Veterinary School surveyed animal care and sheltering facilities across Ohio in 1996 and in 2004. They reported the following trends (Table 3) in the “per 1000 humans” benchmarking statistic for the whole state.

Dog euthanasia decreased from 1996 to 2004.

In general, data from individual states combined with the other data sources employed in this review support the results of national surveys and estimates.

## 5. Responsible Pet Ownership Developments in the United States

Another factor affecting the decline in intake and euthanasia numbers is the cultural shift in how pet owners relate to their pets. Responsible pet ownership and the perception that dogs are part of the family is a concept that has been growing over the last 30 years. In the 1970s, when humane campaigns to address the pet overpopulation issue started in the US, about 25% of the total dog population was estimated to consist of street dogs (roaming the streets where ownership status was unclear—[38]) and millions of dogs and cats were killed in the shelters every year [4]. Today, there are very few “street” dogs and the euthanasia rate of dogs in shelters has fallen by more than 90% even though the total pet dog population has doubled by comparing data (see [4,33]). We argue that this change is one of the indicators that US pet owner relations with their dogs has changed. However, we have to rely entirely on indirect indicators to document the change in human-dog relationship because there are no reliable research reports documenting this change over time (e.g., see [39] as an example of the type of measure that could have reliably demonstrated the change). One possible time-series dataset that could be employed to support the opening sentence conclusion is the biennial survey by the American Pet Products Association [40].

In the past decade, there have been significant changes in the source of pet dogs coming into the home [40] (Figure 12). The percentage of owned dogs that were adopted from shelters and rescues has increased from 15% to over 35% in just ten years (which provides another source supporting the Petpoint trends) while the percentage of pet dogs bred at home has dropped from 5% to under 1% over the same period. If one looks at the percentage of dogs that were clearly acquired purposefully (adopted or purchased) versus acquired serendipitously (from a friend or relative or as a stray), then the percentage acquired purposefully has increased from 46% to 62% in ten years while the proportion acquired serendipitously has decreased from around 37% to 26% over the same time period.

The development of the pet industry also reflects the changing dog-human relationship (see Figure 13). The Bureau of Economic Analysis of the US Department of Commerce [41] produces monthly, quarterly and annual tables of consumer expenditures. The data publically available on the web site (see GDP and Personal Income, Underlying Detail, Section 2 (Personal Consumption Expenditures)) reports expenditures on Pets and Related Products (line 126) and Veterinary and other services for pets (line 225). The graph in Figure 13 shows the relative level of personal expenditures on these two categories compared to total consumer expenditures. The relative amount spent on pets and on veterinary services has increased by 5-fold (pet products) and 3.3-fold (veterinary services) since 1959 but the growth has not been uniform over the last sixty years. There was no relative increase in expenditures on pets from 1975 to 2000 but both veterinary expenditures and spending on pet products has been growing faster than general consumer spending since the beginning of the century. These increases are another indirect measure of a growing attachment to pets in the United States.

Owning a dog has become a conscious choice rather than incidental and with this shift we see a changing relationship. One of the first indicators is the level of confinement of companion dogs (from free roaming to confined and clearly associated with a household). This happened around the same time that sterilization became part of the basic care. Following this change, dogs moved into homes and became identified as more formal members of the family. One possible indicator of this changing relationship is the proportion of dogs sleeping inside at night. (APPA surveys (Figure 14) are more specific in that they ask if dog owners allow their dogs to sleep in their beds and not simply inside.)

Finally, in 2007, Harris [42] conducted a poll of pets as family members in US households and has repeated the poll three times since then (in 2011, 2012 and 2015). In general, the vast majority of pets are viewed as family members, growing from 88% in 2007 to 95% in 2015 (Figure 15) in this survey. Additionally, over 45% buy their dogs a birthday present and 71% share their bed with their dogs, which indicates a strong emotional attachment to their dogs and is higher than the APPA survey results.

## 6. Discussion

Dog management in the United States has evolved considerably over the last 40 years. While programs were devised and implemented in the absence of much data [43,44], the possible effects of interventions may still be tracked. Pet dog and cat sterilization is widely regarded as one of the major reasons for the decline in shelter intake and euthanasia from 1970 onwards, despite the doubling of pet dog and cat populations. We speculate that a combination of factors have markedly decreased shelter intake and euthanasia and these include increased responsible pet ownership behaviors such as sterilization, dog containment, and pet identification. Increased rates of dog sterilization have been facilitated by differential fees for licensing of sterilized dogs, increased availability of low-cost pet sterilization through municipal and animal welfare agencies, high volume specialty spay-neuter veterinary clinics, and incorporation of sterilization as standard veterinary care by private practitioners. Increased levels of pet identification have occurred through licensing compliance and microchipping. Changes in dog-human relationships and increased expenditure on dogs are also likely reflecting a growth in responsible ownership behaviors. In addition, increased numbers of dogs adopted from shelters, and a greater proportion of the owned dog population acquired by adoption, appears to be contributing to decreased euthanasia rates since 2005 [20].

Before 1970, the sterilization of pets by veterinary practices was relatively rare. This apparently changed very rapidly in the 1970s. During the 1970s, there was also a substantial decrease in the shelter intake of dogs in Los Angeles and across the country. An internal and unpublished report by The HSUS looked at shelter intake trends for several hundred shelters in the US during the 1980s and found that a declining intake was associated with differential licensing fees (the owners of intact animals had to pay a higher annual dog license fee). We suspect that, in addition to increased responsible pet ownership behaviors, these differential dog licensing rates combined with changing veterinary practitioner behavior (it has been reported that private practices carry out 80% or more of dog and cat sterilizations annually [45]), contributed to the intake declines in the 1970s and early 1980s.

Shelter animal intake levelled off in the 1980s but dog intake began to decline again in the late 1980s to mid-1990s. We do not know why this occurred but we speculate that another shift in veterinary private practice at the beginning of the 1990s (requiring pet owners to “opt out” of sterilizations as part of responsible puppy care) was one contributor to the reduction in shelter intake from the 1990s onwards. This attitude change toward sterilization in the private veterinary sector and the ongoing expansion of low-cost community sterilization efforts (especially in low-income neighborhoods in recent years), may have sustained the declining trend in shelter intake (and euthanasia). There are likely other factors involved in the decline (such as more responsible dog ownership, including increased containment and identification of dogs through licensing and microchipping) but there have been very few attempts to identify such factors and even fewer attempts to quantify them. Data shows that, across the US, dog (and cat) shelter intake continued to decline despite an increasing pet population.

Today, shelter animal euthanasia is over 10 fold lower than in the 1970s. While declining intake appears to have been strongly associated with declining euthanasia up until 2010 (e.g., [19,46,47,48,49]), an increase in shelter dog adoptions has also become an important driver in the last decade (see Figure 5).

There are still considerable differences between states but the general national trend is clear. The level of control of pet dogs has increased steadily from the 1970s to the present. The proportion of dogs allowed to roam free on the streets is negligible in most communities and a larger proportion of families regard their pet dogs as “family members”.

In summary, campaigns to improve dog owner behavior in the last 40 years have created the changed dynamic we see between humans and dogs. Shelters can focus on adoptions rather than providing humane euthanasia and dog owners have largely adopted a pet care regime that includes sterilization and licensing, and confinement of pet dogs. This progress from relatively uncontrolled to controlled dog population is something we suspect is a trend which occurring globally even in countries with large street dog populations. This review and the US model itself can therefore potentially provide a template for other countries.

## 7. Conclusions

This review has covered a complex mélange of different data sets and has attempted to describe, by referencing these data, what has happened to dog management in the United States from around 1970 to the present. There have been a few attempts to compile a scholarly review of all the data (e.g., Marsh, 2012 [45]) in one place, but the field is hampered by the lack of an accurate data set describing the US animal shelter world over the past fifty years. There have been very few attempts to undertake a scholarly examination of national trends, probably because of the lack of what would be considered “reliable” data. We have chosen to use most of the available data (with some exceptions) to construct an overall view of the trends in dog shelter intake and euthanasia from 1970 to the present in the USA. We agree with Marsh (2012) [45] that the dominant influence on shelter euthanasia from 1970 to 2010 was the declining intake of dogs into shelters in the USA. As intake declined, so did dog euthanasia rates. However, this changed around 2010 (or perhaps earlier—around 2005—see Figure 6) when shelter adoptions of dogs diverged from the intake trend. This may have been assisted by the 2009 Ad Council campaign to increase shelter dog adoptions (still ongoing). In the absence of careful data collection and research, we will not be able to identify what is driving current trends.

## Figures and Tables

**Figure 1 animals-08-00068-f001:**
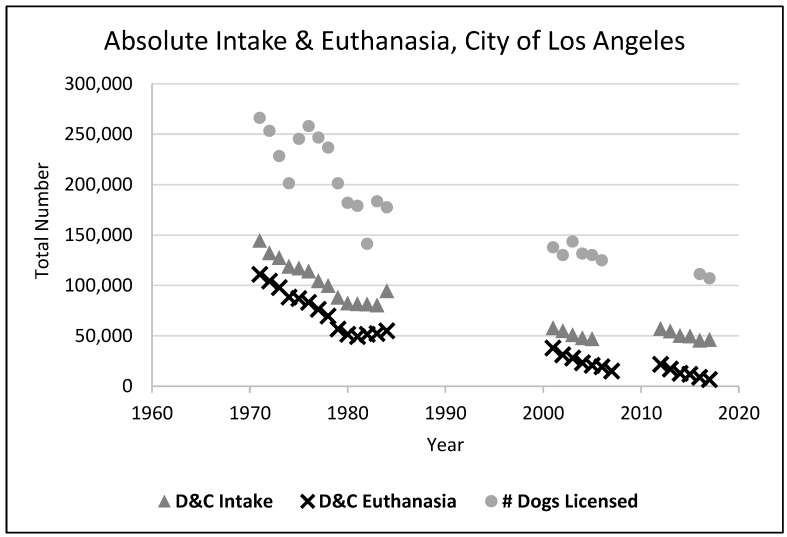
Intake and Euthanasia of Dogs and Cats by Los Angeles Animal Services [10,13,14,15,16].

**Figure 2 animals-08-00068-f002:**
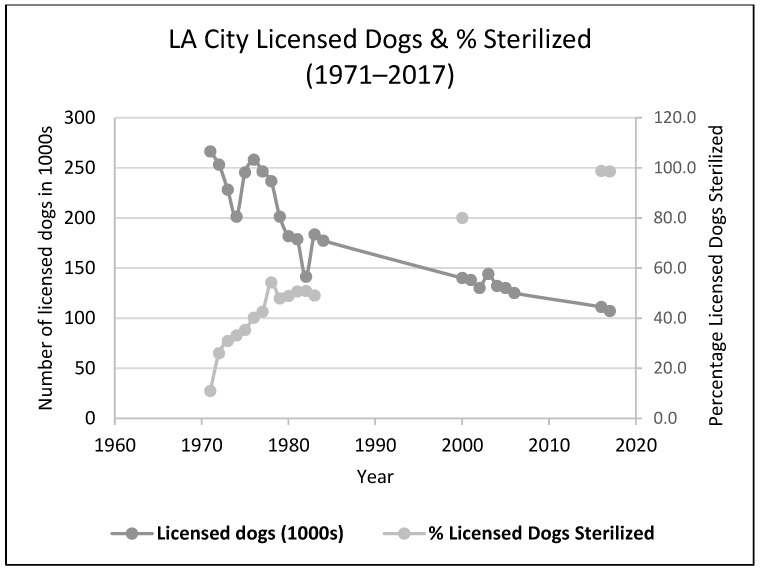
Licensed dogs in the City of Los Angeles and the Percentage Sterilized [10,13,14,15,16] 2000 data point from McKee, 2000 [17].

**Figure 3 animals-08-00068-f003:**
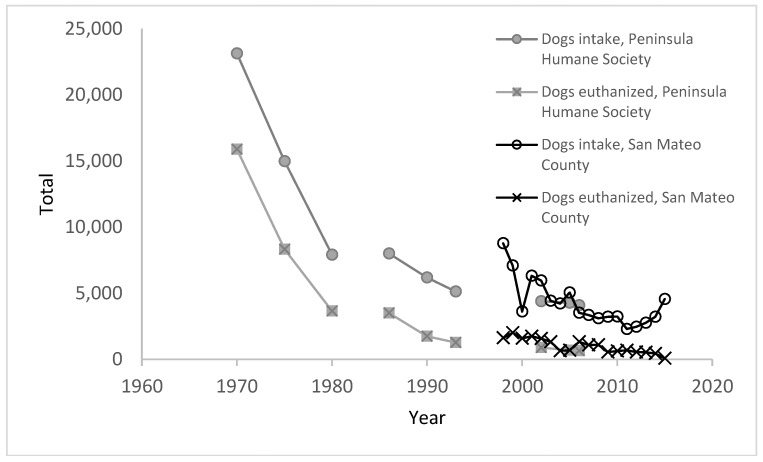
Change in dog intake and euthanasia at Peninsula Humane Society (the main shelter in San Mateo County, CA, USA) from 1970 to 2006 and in San Mateo County between 1997 to 2015 [13].

**Figure 4 animals-08-00068-f004:**
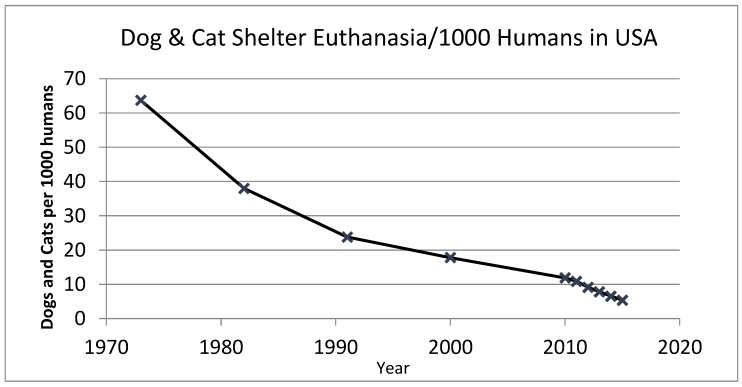
Figures for the graph below are based on rough estimates of the number of dogs and cats euthanized per 1000 people in shelters in the USA [4,25]. The more recent estimates are supported by more robust raw data sets drawn from Clifton (2014) [26] and PetPoint sheltering reports [27].

**Figure 5 animals-08-00068-f005:**
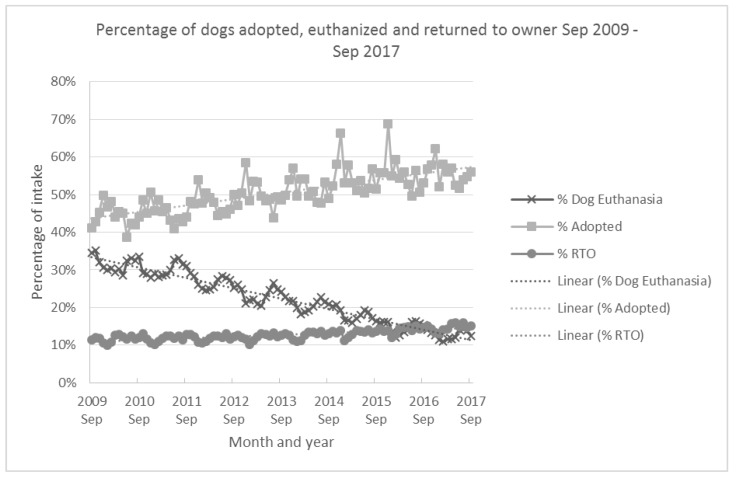
Percentage of dogs adopted, RTO (Returned to Owner) and euthanized (of total dog intake) nationwide based on PetPoint data (from 900–1200 shelters and rescue organization) September 2009–September 2017 [27].

**Figure 6 animals-08-00068-f006:**
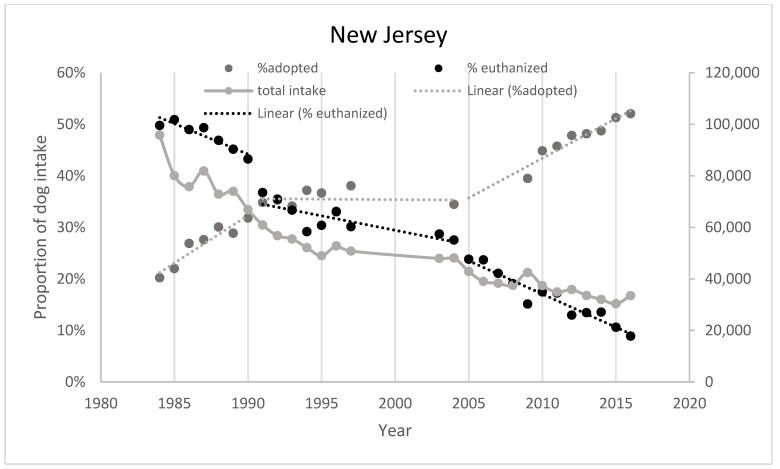
Shelter dog adoptions and euthanasia as proportions of intake as well as total dog intake, based on data provided by the State of New Jersey.

**Figure 7 animals-08-00068-f007:**
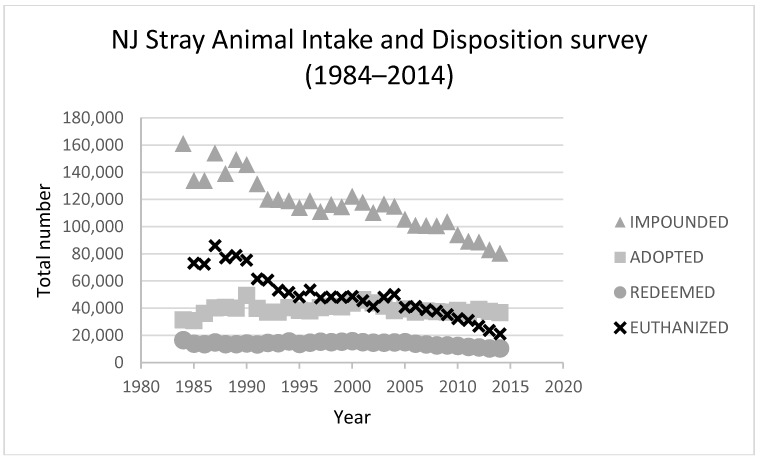
New Jersey Stray cat and dogs survey results 1984 to 2014.

**Figure 8 animals-08-00068-f008:**
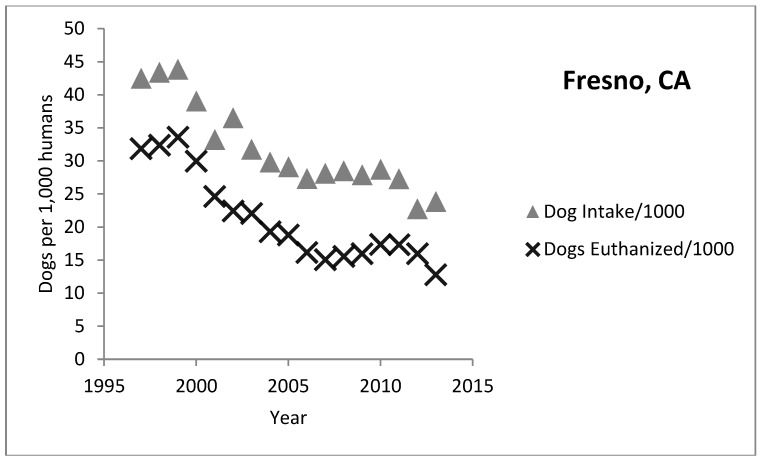
Fresno, California intake and euthanasia per 1000 people.

**Figure 9 animals-08-00068-f009:**
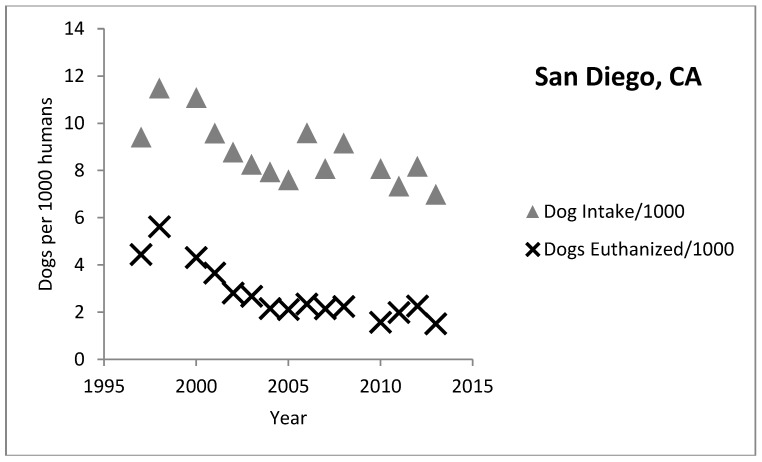
San Diego, California intake and euthanasia per 1000 people.

**Figure 10 animals-08-00068-f010:**
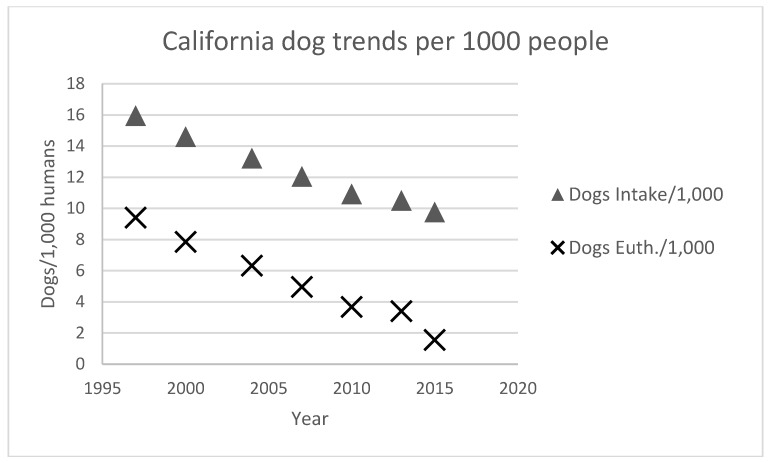
California dog shelter trends per 1000 people living in California.

**Figure 11 animals-08-00068-f011:**
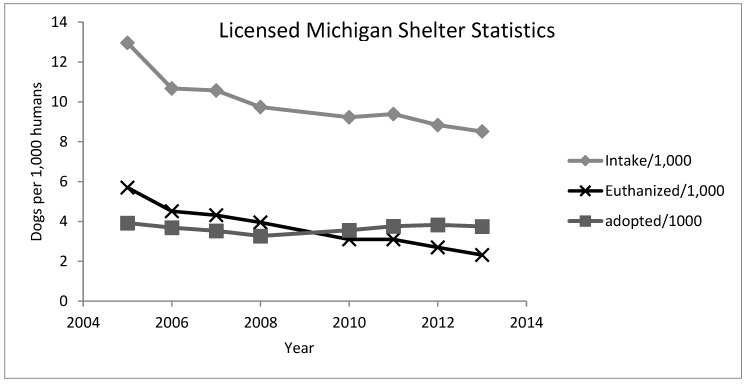
Intake and euthanasia per 1000 people living in Michigan (raw data retrieved from: [26]).

**Figure 12 animals-08-00068-f012:**
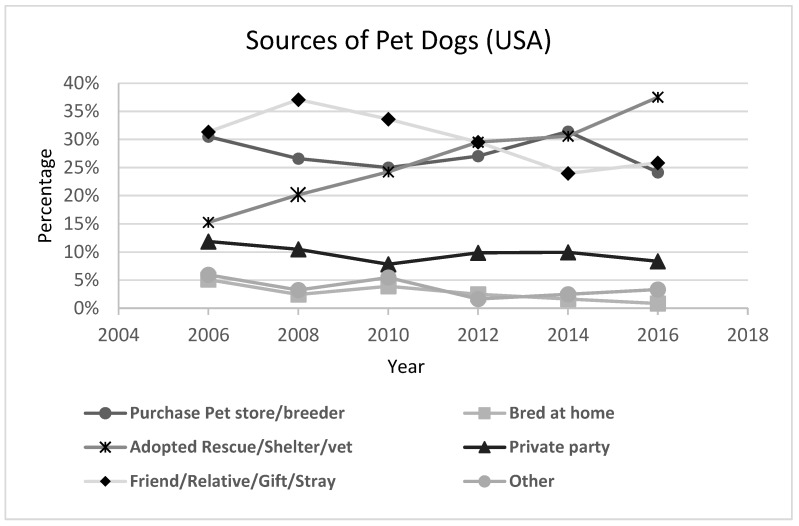
Acquisition of dogs in the United States [40].

**Figure 13 animals-08-00068-f013:**
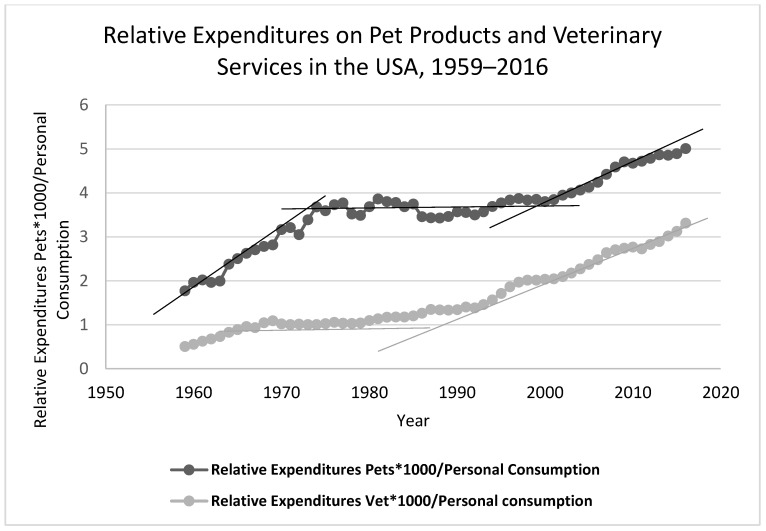
Expenditures related to dog keeping and veterinary care [39] (Note: the straight lines in the graph are not calculated trend lines but are included to distinguish the different periods of relative expenditure growth).

**Figure 14 animals-08-00068-f014:**
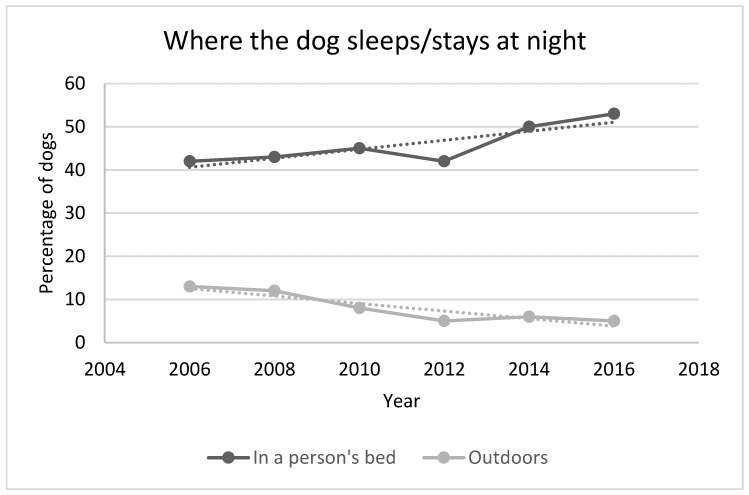
Where dog owners keep their dogs at night and where they sleep [38].

**Figure 15 animals-08-00068-f015:**
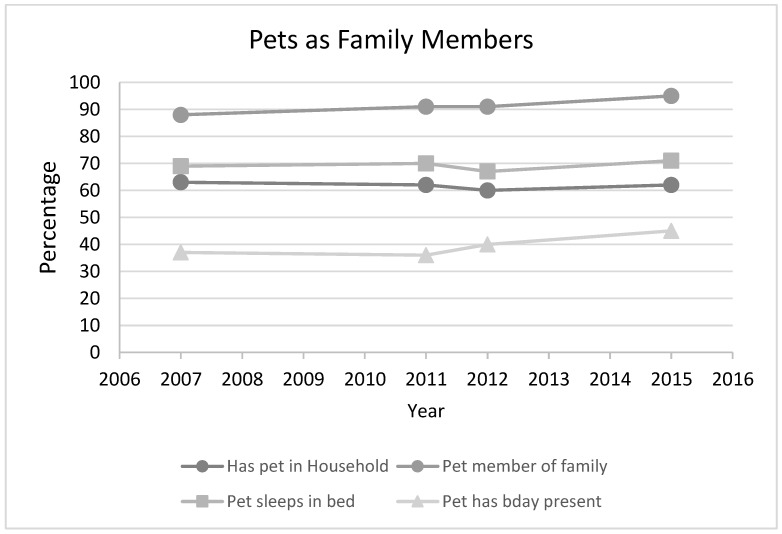
Pets as family members.

**Table 1 animals-08-00068-t001:** Calculations based on data presented by Woodruff & Smith at the 2017 North American eterinary Conference in Florida and PetPoint data indexed for 1000 organizations.

Topics	Woodruff & Smith (2016)			PetPoint (2016)
**Number**	Total Number	Lower 95%	Upper 95%	
**Shelters**	7076	6399	7890	NA
**Total Dogs Entering**	5,532,904	5,003,528	6,169,579	4,171,017
**Adopted Dogs**	2,628,112	2,376,660	2,930,531	2,302,829
**Dogs Returned to Owner**	969,443	876,689	1,080,998	591,375
**Dogs Transferred**	778,385	703,911	867,955	642,856
**Dogs Euthanized**	776,970	702,631	866,366	592,255

**Table 2 animals-08-00068-t002:** Number of dogs euthanized per 1000 people were calculated based per State where official numbers were reported (raw data retrieved from [26,32,33]).

State	Year	Dogs Euthanized in Shelters	Human Population (2010)	Dogs/1000 People (2011)	Dogs Euthanized per 1000 People	% of Pet Dogs in State Euthanized in Shelters
California	2011	176,907	37,253,956	177	4.69	2.65%
Colorado	2013	6968	5,029,196	264	1.36	0.52%
Delaware	2011	2012	897,934	180	2.22	1.23%
Maine	2012	644	1,328,361	226	0.48	0.21%
Maryland	2011	10,477	5,773,552	157	1.8	1.15%
Michigan	2013	22,909	9,883,640	206	2.32	1.13%
Nevada	2011	14,679	2,700,551	212	5.39	2.54%
New Hampshire	2012	346	1,316,470	161	0.26	0.16%
New Jersey	2011	6023	8,791,894	152	0.68	0.45%
North Carolina	2013	62,269	9,535,483	261	6.45	2.47%
Virginia	2013	16,519	8,001,024	210	2.04	0.97%
Total for 11 States		296,867	90,512,061		3.28	
Estimated USA Totals	2010	1,723,039	309,350,000	225	5.57	2.48%
(from PetPoint & AVMA data)						

**Table 3 animals-08-00068-t003:** Ohio survey results.

Topic	*1996*	*2004*
Dogs Handled/1000 people	19.14	15.59
Dogs Euthanized/1000 people	11.50	6.85
Total Animals Handled/1000 people	29.41	26.84
Total Animals Euthanized/1000 people	18.73	14.89

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
