# Peer review of "Dog Population & Dog Sheltering Trends in the United States of America"

_animals, 2018, doi:10.3390/ani8050068_

Round 1

Reviewer 1 Report

see attachment

Author Response

Responses to Reviewer Comments by Rowan and Kartal

Reviewer

R3  L42

Add overview of US dog   legislation

We do not believe such an   addition is needed for this review and it would be awkward to include here.

R2  L48

Says Google search items   have little to do with shelters.

We disagree – following   sentences clarify but we have added citation to recent paper that tracks   scientific publications on pet demographics.

R3  L60

Delete “in our opinion”

Done

R3  L61

Add section on data   retrieval and analysis

Done

L63

Added section on data   sourcing and data reliability

L91-122

We agree that more detail   is necessary for the data sources but have now placed it in a Supplementary   Material file as well as some of the charts that some reviewers found   overdone. 

R4 L91-122

Petpoint data not   representative of national trends.

We disagree with the   reviewer.   As we review data from a   variety of different sources, it is clear that shelter animal intake is   plunging across the country.  This is   what the Petpoint data also indicate.     If R4 could provide some countervailing data (and more than just the   cat data from Colorado), then we would agree that stronger support might be   needed.

R3 & R4 L138

Both wanted a source for   the 3,500 shelters. 

ANR has been developing lists   of animal shelters since the 1990s and organized the survey leading to the   current list of 3,350 (dating back to 2008 – hence the increase to 3,500   today). This is the most accurate shelter list available in the USA.  It does not include animal rescues which   have exploded in number in the last decade but which still account for no   more than 10-15% of total intake and outcomes. 

R4 L145

Author’s opinion – no data.

There is data to support   this (in 1985, open admission shelters were spending an average of $100-140   per animal handled but today they are spending an average of $500+).  Closed-admission shelters are spending   $1,500 to $4,000 per animal.  However,   the data has not been systematically collected. 

R2 L157

Info on speed of adoption   of ordinances

Nothing systematic has been   collected so no info to offer here. 

R1 – L164

Low-cost

We believe “low-cost” is   well understood and have not changed.    And it does not necessarily mean subsidized.  High-volume clinics can typically sterilize   dogs and cats for a fraction of private clinics who perform a few   sterilizations a week. 

R4 L168

Los Angeles sterilization   rates

The mandatory s/n ordinance   went into effect in 2008, well after the huge increase in sterilization   rates.   The HSUS Pets for Life program   has been active in south LA (a low income area) where most companion animals   are neither licensed nor sterilized (according to personal comments from PfL   manager. 

R4 L173

Unsupported

Have been unable to put   hands on material supporting this. It has been removed.

R4  L184

Include Zawistowski et all   1998

Done

R2 L204

Meaning of “Perforce”

It is a perfectly good word   meaning “of necessity; necessarily; by force of   circumstance”

R4 L205-7

Make better argument

We have included a   right-axis on Figure 5 to show the growth in pet dog and cat populations over   the same period to connect the total population with shelter euthanasia   trends.

R4  L214

Cat reference removed

R4 L 219

Title of Fig 3 changed.

R4 L225

Where is AHA data &   National Council

AHA data is badly flawed so   is not included and National Council never produced a considered estimate of   national intake and euthanasia or trends.    

R2 L230-5

Petpoint discussion

Move to suppl material

R4 L237

Clifton data

See suppl material

R3  L 237

Regression analysis

We could easily provide   such analysis but does anyone honestly believe it would make the chart any   less convincing? 

R4 L245

Update on 1998

None available.  We do not believe necessary.  Dropped “now” and added “in 1998”. 

R2 L250

Source requested.

Done

R3 L252

Decrease in euthanasia due   to no-kill?

No, it is not.

R4 L253

Data to support adoptions

Presented in Petpoint   charts – revised.

R2 L253

Euthanasia numbers

Yes but also on other data   sources.

    L267

Charts have been changed

R1 L271

Relinquishment is real   issue

This misreads the situation   at many shelters in the country – about 40% of the dogs entering are stray,   another 40% are “owner releases”. 

R3

Chi-square analysis

Not sure that regression   analyses add much – the chart indicates a trend over time.

R4 L306

Define a shelter and   address rescues

Done

L308-320

Moved this discussion to   Suppl Materials

R2 L321

“Editorializing”

Authors are supposed to   present arguments and conclusions.  Not   sure what R2 means by “editorializing” – suspect the argument is something   that s/he does not agree with. 

R4 L325

Considers data sources   substantially different

That is certainly R4’s   prerogative. 

R2 L330

Removed section in parens.

R1 L345

Density issue

We decided not to include

R2 R4 L359

Source of Clifton data is   an issue. 

This is something we   address in more detail in Suppl Material.   

R4 L367

Source?

The source is the AVMA data   cited in the para.

R1 L401

Refer back to LA 1970 data

This is a good point.  We should do so.  But we cannot compare the 1970 data with   the current data because the Rush data is for one entity and there are   several large entities in LA county. 

R1 L412

Unexplained markings

Corrected

R4 L426

R4 argues that associations   are not causations.  True

We do not state that one   trend causes the other.  We just   present the data and let the reader make his/her own inferences.

R2 L427

We have omitted Ohio.

R2 L453

Too speculative

We disagree.

R2, R4 L469

Want a reference

We do not believe a   reference is necessary.  This is the   opening sentence of a section in which we provide evidence of a changing   relationship.

R4 L527

Sources

Done

R4 L529

Need an explanation

The text provides a description   of the trends but an explanation would require further research on changing   human-dog bond in the USA.  We prefer   not to speculate!!

R4 L537

Wants National Council data

We do not believe the   National Council on Pet Population provides much in the way of trend   data.  There were a few useful studies   that examined data from a few shelters at a specific time but their surveys   of shelters used flawed methodology and does not provide much in the way of   insight about the reasons for the trends.

R4 L544

Criticism of lack of data

We have rewritten this   section to address the data lack.  

Reviewer 2 Report

It is commonly discussed in animal welfare circles that dog population management has drastically changed in a relatively short amount of time, yet the details regarding how and when incremental changes have taken place over the past 40+ years have not been provided in a single, peer reviewed manuscript. The authors take on this admirable challenge in this manuscript.

I do have many concerns about how this information is presented, however. The paper is listed as a review, yet I think it would help to have a Methods section describing in detail the types of sources (letters, websites, surveys, academic sources, etc.) the authors used to piece together the history of dog population management and the strategies they employed to locate this information. In general restructuring the paper would bring clarity to the authors' main points and help the reader understand what sources and data support those main points.

Another major concern is that there was consistently a lack of clarity regarding the source of information. For example, what is the source for "This number had grown to around 3,500 shelters in 3,100 counties by 2015" (lines 60-61)? Similarly, I don't see a citation for the 1973 survey described in line 62.

Line 44: The fact that 12,600 items resulted from a Google Scholar search  that included the term "animal shelters" does not provide convincing evidence that numerous articles have been published about animal shelter trends. Papers that included this term would not necessarily have focused on on animal shelter trends.

Lines 78-79: Can the authors provide information about the speed at which local municipalities adopted licensing ordinances and began supporting sterilization campaigns?

Lines 91-85: From what source did the 50% and 100% data come? Similarly, from what records did the "10,000 sterilizations a year" come?

Lines 96-98: A citation is needed for the statement regarding how veterinarians viewed sterilization surgeries prior to the 1970s.

The paper includes more graphs than necessary, and the formatting of the graphs varies greatly. In most graphs, the y-axis is missing a label. Figure 2 should not include count data and percentage data on the same axis. Figure 5 is very busy and difficult to interpret. For some graphs, the different series are represented by different shapes; for other graphs, different series are represented by different colors of a single shape. When printed in black and white, the series that have the same shape all look identical. Why do some graphs include equations and R2 values, yet others don't? Some graphs include best fit lines; others connect the dots; others do both; and yet others do neither.

Line 127: I'm not certain what was meant by "perforce."

Lines 128-129: "Today, most observers are comfortable concluding..." Can the authors cite evidence that supports this?

Lines 144-155: This paragraph provides a clear description about how some of the information was sourced, as well as some of the limitations inherent in the sources. It seems that this information and some additional information could come right after the Introduction and would add clarity to the ways the information reported after the Introduction was compiled. More detail about the PetPoint data would be helpful. For example, does the PetPoint dataset include data from all 50 states? Is it biased toward particular states?

Line 171: What is the source for the information about New Jersey and Massachusetts?

Lines 174-180: Is this section based on PetPoint data? 

Lines 184-186: Be more explicit about how the data "underlines the effect adoptions have on euthanasia." Are these variables significantly and negatively correlated?

Lines 189-195: Who is making the assumption that the PetPoint data represents 20% of all shelter and rescue operations int he US? The authors? PetPoint? Clarification is needed. Earlier in the manuscript, the authors mentioned that municipal shelter data is underrepresented in the PetPoint data, and so it seems that some of the extrapolations drawn from the PetPoint data to the entire shelter and rescue population could be problematic (e.g. may underestimate intake numbers since municipal shelters handle more intakes that rescues, which may be more heavily represented i the PetpPoint dataset).

Lines 194-195: Walk the readers through how you came to the conclusion that "intake numbers are less important influences on euthanasia numbers."

Line 212: The authors are editorializing when stating that "we believe that..." Let the information speak for itself. Furthermore, there do seem to be some major discrepancies between the PetPoint and Woodruff and Smith datasets. The authors point this out, yet report that one dataset indicates that the other is valid.

Line 218: Provide a citation (web address) for Shelter Animals Count.

Line 220: Why is there so much information about Florida in the parentheses? Were most of the data Shelter Animals Count compiled from Florida? Is Florida just being used as an example?

Line 233: "Relatively good" sounds like editorializing.

Line 237-245: What is the source for the information reported here? Is the information from the PetPoint dataset? This is unclear.

How is Table 2 organized? States do not seem to be grouped by region,and they are not alphabetized. They are not arranged by year, nor by number of dogs euthanized nor human population.

Lines 250-256: Cite the AVMA sourcebooks.

Line 267-268: The statement in brackets seems tangential.

Line 272: Provide some information about what is in this section and why the specific states within it were selected.

Lines 289-299: A citation is needed for the California data (and for the data from the other states in this section). Is there a citation to support the speculative comment about immigrant laborers and dog ownership?

Line 315: Why were Michigan and Ohio combined within a single section even though the first paragraph focused on Michigan and the second on Ohio?

Lines 330-331: The statement in parentheses is too speculative.

Lines 337-340: Sources are needed to support the arguments about cultural shifts.

Lines 346-353: What is the source of this information?

Lines 368-369: Insert "likely" between "also" and "indicate."

Line 379: Provide a citation for the APPA data.

Lines 393-397: While this study was interesting, it seems tangential to the point of this paper.

Lines 415-417: The paper does not clearly spell out how the authors established that euthanasia rates have decreased 10-fold or that adoptions have become "an important driver..."

Lines 421-422: The paper is focused on the US. The last sentence seems tangential to the point of the paper.

Author Response

(The authors gave the same response as above.)

Reviewer 3 Report

General comments

The authors are reporting dog population changes from the Seventies until present days, in the number of animals entering shelters and being euthanized, and suggesting that these changes are likely due to a shift in the veterinary care practices, especially an increase in sterilisation procedures among other cultural changes.

The data are interesting and well presented, however, I’m not sure if this work should be presented as a scientific review. Because of the lack of published scientific research, the authors collected data from available databases and reports, and it is clear that a lot of effort has gone into retrieving a large amount of data. The authors are presenting unpublished data and presenting descriptive analysis of these data. In my opinion this work would be more appropriate as a retrospective study than a scientific review. I the case of a retrospective study, I would suggest to perform some simple statistical analysis (see below) to support their statement (e.g. regression models to confirm a significant difference in population change in time)

Further, I think the authors are missing some key elements proper of a review: they should state how is this review beneficial to the scientific community? what are the gaps of knowledge that they found that should be covered by scientific research? What can we learn from your report that can help the scientific community to move forward? These are all thing that should be included in their discussion.

Detailed comments:

Simple summary and abstract: I think both summary and abstract should state more clearly what the aim of the review is, why it is needed and how it will benefit the scientific community

Introduction: I think a brief overview on US legislation on euthanasia for shelter dogs and differences at State level should be presented in the introduction.

L53 delete ‘in our opinion’

Methods: a paragraph describing how the data and literature was retrieved and the approach used should be added.

L61 This number…..by 2015. Add reference

L156 here and elsewhere linear regression analysis can confirm if those population shifts are statistically significant or not.

L187 chi square test could reveal if these increase in adoption and decrease in euthanasia are significantly different of what could be expected by chance.

L194 please describe what PetPoint is and how data are fed into the system.

Ll 173 could the decreases in euthanasia in recent national trends be also due to an increase in the number of ‘no-kill’ policies in shelters? I’m not familiar with the situation in the US, but it seems they are never mentioned.

Ll192-193 this is why it’s important to know how data are fed into the system. If data are added by shelters on a voluntary base, these data could be biased toward ‘good’ shelters that make more effort to increase adoptions and decrease euthanasia, not being representative of US nationwide.

L220 what is the difference between shelter and rescue in the US? It might not be the same around the world.

Figures: although figures are helpful for the reader and help visualise the date, I feel here there are too many, some could be merged into one and others are probably superfluous and could be substituted with a description. Also, a lot of figures need axis labels and where possible measurement scales should use the same units to ease comparisons. Finally, I would try and standardise the layout of the graphs, some have squares in the background, some have lines, some have plain white some have grey, some have legend in a text box with border, some not etc…

Here some suggestions:

Fig 1 & 2: use same scale unit on y axis, maybe these figures could be merged into 1

Figs 4, 8, 9, 11 ,12, 14, 16 add axis labels.

Figs 11,12,13 check legend, there seem to be ideograms which I don’t think are intentional

Figs 17,18 are maybe not so relevant and could be substituted with a description

Discussion

This need more work, the authors are presenting here a summary of their findings but are not really discussing their findings. Also see my general comments.

Author Response

(The authors gave the same response as above.)

Reviewer 4 Report

This manuscript covers an important topic that has not been updated in some time.  My primary concerns are the lack of rigor about the data sources and their potential biases as well as a lack of organization and consistency.  There are also a few references that should be included.  Most of the data presented comprise an ecological study where group data is used to make inferences about individuals.  This should be acknowledged and the limitations discussed.

I have specific comments below.

This manuscript isn’t really about “population management” but rather about dog population trends so the title could be more accurate.

Line 14: I don’t believe that the authors have made the case convincingly that sterilization was the driver. I don’t disagree but I think the argument needs to be tightened up and the limitations of the data acknowledged.

Line 17 and line 30: similarly the data tends to really support the idea that decreasing intake is a key driver of decreasing euthanasia, the regional and state data support that conclusion.  Again, increasing adoptions is likely a critical element of the puzzle but I might argue (and some of the information in sections 3 and 4 would support it) that the decline in intake provide shelters with the breathing room to be able to improve their adoptions programs through open adoptions, keep animals healthier and therefore able to be adopted and thereby decrease euthanasia.

Line 61: where is the reference for this estimate?  That would be helpful.

Line 67-8: this reference is to that author’s statement of opinion.  No data is provided.

Line 91: sterilization is mandatory in LA so this finding is not surprising (similarly for figure 2).  Please add this information.  However, the questions remain: how many unlicensed dogs are there?  And what is their sterilization status?  And furthermore, what percentage of dogs entering the shelter are intact?  Please address these questions.  Then when using licensing data to estimate the private practice involvement, clarify that this is only a segment of the total dog population.

Line 96: is there nothing in the veterinary literature about the role of private practice veterinarians and sterilization to support this sentence?  The parallel shifts in veterinary medicine seem important for understanding the cultural shifts discussed later in the manuscript.  Consider Valuing Animals: veterinarians and their patients in modern America chapter 5 by Susan D. Jones.  Figure 1 illustrates that licensing numbers, while parallel to shelter statistics, are clearly reflecting a different population of animals.

Top of page 4: consider also Zawistowski et al. JAAWS 1998 for trends.  And if there is actual data for the number of sterilizations, please report that primary information.  This is such an important point to be relying on a secondary source.  It is interesting that reference 5 also states that sterilization isn’t a remedy for many of the factors that increase risk of surrender to shelters.

The intermittent inclusion of cats in the data is a bit confusing.  I’m assuming that has happened because the data were not broken out by species?  Please orient the reader and explain this as appropriate since this manuscript is about dogs.  Line 136-40: I don’t see how this is relevant to the dog situation that is the focus of this manuscript.

Line 128-9: I agree. And I think that the argument about how sterilization influences euthanasia hasn’t been clearly stated.  I would like to see this section begin by clearly stating what the authors believe is the situation and then marshalling their data and interpretation to support that situation.  It isn’t very clear or compelling as it currently stands.  And the argument for differential licensing really hasn’t been made at all.  Furthermore I’m a bit confused about the focus of the manuscript: the introduction states it is both shelters and US pet population trends.  I think that separating those two populations in the sections in the manuscript would be helpful.  That is where, perhaps, the differential licensing comes into play.  But mostly, it seems that the shelter data is the focus of all but the last section.  Please consider how to make the information about these two populations flow seamlessly through the manuscript.

Figure 3: Is this figure mislabeled?  If the euthanasia is overall then I think it makes more sense to show overall intake data.

Line 145-6: there is also AHA data in the 1980s and National Council Data in the 1990’s which could be considered here. This paragraph is where some serious discussion of the strengths and limitations of these data are needed.  How many shelters were assumed to exist to use the PetPoint data and extrapolate (I’m assuming that is what the normalization means?  Please clarify)?  Are shelters and rescues used here (I think so) so that difference between shelters and rescues in other data sources needs to be made throughout and definitions provided.

Line 158: Clifton’s data: can you share where those data come from?  And curious that this reference also says that we can’t adopt our way out of euthanasia.  Please address later on.

Line 165-6: this is based on 1998 data. Please clarify or update.

Line 174:  This statement is what the authors have proposed. Please clarify that and then provide the data to support this.  I think again that there is a gap in the argument (that I have outlined above) and which would allow the authors to explain why we aren’t “adopting our way out of euthanasia” but rather able to redirect resources to get animals out of shelters alive. 

Figure 5: there is a “dog outcomes” is that the sum of adoptions and euthanasia?  I think that dog intake would be a clearer data line and transition better from the previous shelter data figures and discussion.  Again a brief phrase in the figure legend explaining that this represents more non-municipal shelters (which might be expected to be larger, have fewer resources and lower adoption rates/high euthanasia rates) and is extrapolated from 1000 US shelters.

Lines 190-2: I am not convinced that PetPoint data is representative of all US shelters due to the limitations of some shelters not having PetPoint or any software and the lack of representativeness of the municipal shelters.  I think it is fair to say that PetPoint shelters show a clear trend but a much better argument would need to be made that these data would hold true if all shelters were included.

Line 194-5: I might suggest that there could be a threshold effect: once a certain level of sterilization has been in place long enough, there are other factors that influence euthanasia numbers.  Adoptions could certainly be one of those factors.  Given that some shelters who are experiencing major declines in dog intakes and are now only seeing dogs with serious medical and behavioral problems, we may be looking at another shift towards increasing euthanasia in those locations.  Something to consider.

Line 197: please describe the normalization process the first time it is mentioned.

Figure 7: seems like this would be more helpful if referenced against intake as the title suggests.  And the adoption data isn’t showing much of an uptick, which doesn’t really support the authors’ arguments.

Line 205-7: how are shelters defined in each of these sources?  There are many different definitions and rescues are sometimes included in the manuscript.  Please clarify.

Table 1: the two data sources seem to have some substantial differences in intake, RTO and euthanasia.  I think discussing why those differences exist could help make the data sources more believable.

Table 2 and discussion about it: where is Clifton getting his data? What are the response rates? 

Line 256: where is the data for this statement?

Line 297-9: please reference this statement.

Figures 9-13: support the argument that declining intake is associated with decreasing euthanasia.  And associations are not causation. The authors would need to explore other reasons and evidence to make an argument for causation.

Figures 11-13: what are the data sources?

Line 333: so 3 states where data is likely to be included in the national statistics have similar trends?  Make a better argument here.

Line 338-9: please reference this statement.

Figure 17: where is sleeps inside apart from in the bed?

The trends in Figure 18 are fairly slight…I think they are real but need some explanation.

Line 402-3: there hasn’t been any data on private practitioners presented.  Please include some or edit.

Line 405-6: I suspect there were other factors operating at the same time.  If the authors discuss them and share how they might or might not interact with these factors, the argument would be much stronger.

I think that including some of the NC data as an example of one way that was used to estimate shelter intake would be appropriate in this manuscript.

Reference 8: please add the subtitle.

Author Response

(The authors gave the same response as above.)

Round 2

Reviewer 2 Report

The manuscript has been substantially improved, and the authors have added citations to make clearer the sources of their information. The newly added section about Data Sourcing helps place the information reported into context. I have a few relatively minor comments.

Lines 214-215: The analyses provided in this paragraph are a nice addition; however, it is unclear what relationship is being described in this sentence and whether the association is positive or negative.

Line 225: What is the basis for the assumption that “Petpoint data cover about 20% of all  shelter and rescue operations in the United States”?

Line 229: One limitation of the Woodruff and Smith study is that they did not account for dogs being counted twice in situations in which they were transferred from one shelter to another. As a result, multiple shelters may have included some of the same dogs in the counts they provided Woodruff and Smith, and thus the estimate is likely to be on the high side. This may be a limitation of other estimates of shelter intakes as well.

Line 278: I suggest introducing this section with a rationale for how the authors chose which states to feature.

Figures 1,2, 4, 6, 8, 9, 11, 12, 13 needs axis labels. In addition, the graphs require reformatting (e.g. fewer decimal places on the y-axis on some graph).

The additions to section 5 have improved this section.

Author Response

please find the responses in the attachment.

Reviewer 4 Report

Overall, the manuscript is still clunky and lacks clear focus.  The title states the paper is about dog population management, the goal states the paper is about shelter demographics and US pet population trends and the data includes both cats and dogs without apparent rhyme or reason.

The authors have not yet succeeded in adding nuance to manuscript about the trends and patterns and the caveats about the sources.  They make statements that are definitive and use data that generally support those statements without considering alternative reasons for the trends. Given that the authors have substantially ignored my suggestions and requests in the first 1/3 of my previous review I am not going to repeat a detailed analysis with specific suggestions.  I require that the authors do that. 

The additions to the manuscript are helpful in providing context.  The supplemental information about the limitations of pet point and how the standardized 1000 shelters (this is still not clear) are done should be in the main body of the manuscript to make these limitations clear in the interpretation of these results (add to paragraph starting on line 71).

Los Angeles also has active enforcement of the mandatory s/n law and the jump in % sterilization between mid-1980 and 2000 could be due to many things, including the obvious decline in the number of licensed dogs…only people who could afford and knew to get sterilization also got licenses. One example where alternative interpretations are ignored.

Interesting that the authors have high standards for the quality of data of some of the published work which makes it unusable but Clifton’s convenience samples which seem to be based on who he knows and which are not duplicated or peer reviewed anywhere are considered to be accurate data.   Similarly Marsh’s non-peer reviewed opinions are given substantial weight and credibility.

This is an important topic and much of the data used is really all there is. However, the authors’ defiant tone in the response to reviewers and refusal to adjust the tone in the manuscript to make it more scientific and less opinion based continues to frustrate me.

Author Response

(The authors gave the same response as above.)

Round 3

Reviewer 4 Report

I really appreciate the authors’ commitment to clearly articulating the sources and limitation of the data for the reader.  This has made it much clearer what the driving trends are and what data are supporting those as well as how strong the conclusions are likely to be.  The manuscript is substantially improved.

Use of the term “responsible pet ownership”.  This term has many different meanings to different people.  Can you please either use a different phrase or define it in the authors’ opinions.  It is in the discussion but is used in the beginning of the manuscript.

Line 107: systematic or systemic?

Figure 1: if the number of licensed dogs is down, does that mean there were fewer owned dogs and therefore that led to decreasing the intake?  I’m not sure of the message for Figure 1.

Lines 188-96: these data are from an original reference cited by Marsh and are based on assumptions without any real data. Please indicate the assumptions on which these estimates are based.  We really don’t know any of these figures.

Please check the legend in Figure 6 for some odd characters.

I’m still a bit confused about why cat data appears (separately from dogs) in some places but not others. For example, cats are listed in table 3 but not shown in a graph similar to that in Figure 12.  Please clarify and adjust.

Figure 12: the title should be sources of dogs, not pets for clarity.

I think that line 463 and line 521 are missing “)”.

Is Figure 15 only dogs?  Or all pets? Please clarify in the text, title and figure.

Author Response

COMMENTS FOR REVIEWER AN SPECIAL ISSUE EDITOR  (Author comments are in blue.)

Use of the term “responsible pet ownership”.  This term has many different meanings to different people.  Can you please either use a different phrase or define it in the authors’ opinions.  It is in the discussion but is used in the beginning of the manuscript.

There have been many changes made in the manuscript to accommodate the concerns of the reviewers and Dr Rand.  “Responsible dog ownership” is not a term that I particularly care for but it is widespread within the US shelter community (and was promoted as one possible answer to the discarding of animals by the National Animal Control Association – that was our first use of the term in the original submission).  We used the term another three times in the original submission - again without definition but referring to the fact that it had come into vogue in US sheltering community since the 1980s.  In this latest version of the manuscript, it is now used 10 times (after encouragement by Dr Rand that a change in owner behavior could also be a cause in the decline in shelter intake).   I am not comfortable trying to define what it means - especially at this late point in the manuscript review.   Most readers will think they know what it means.

Line 107: systematic or systemic?

Systemic is correct.

Figure 1: if the number of licensed dogs is down, does that mean there were fewer owned dogs and therefore that led to decreasing the intake?  I’m not sure of the message for Figure 1. 

There is no specific message as regards dog licensing.   Dog licensing is a moving target in US communities with somewhere between 25-50% of owned dogs being licensed.  In cities with large low income areas, licensing tends to be on the low side.   We suspect the number of licensed dogs has declined due to changes in city demographics and varying intensity of effort by the Department of Animal Services to collect license income.  We did not comment on the licensing numbers because there is no independent survey of pet ownership in Los Angeles over this period to provide a sense of any change in dog ownership.   

Lines 188-96: these data are from an original reference cited by Marsh and are based on assumptions without any real data. Please indicate the assumptions on which these estimates are based. 

We really don’t know any of these figures.   We are simply reporting Marsh's claims here.  Like the reviewer, we do not know what Marsh based his claims on.  However, we do have estimates of sterilization rates in Los Angeles in the 1970s (see table below).    Certainly, in 1972 and 1973, over 90% of the sterilizations must have been performed by private practitioners (even without noting that the number of pet dogs was probably 2-3 times the number of licensed dogs and one presumes that at least some of the unlicensed dogs were being sterilized as well.   By 1975, one would have to start correcting for deaths of sterilized dogs to obtain a rough estimate of the percentage of sterilization being performed by the municipal clinic.  One would also need to account for cat sterilizations by the municipal clinic.   As the reviewer has noted, the data are not particularly robust.   We would have preferred something more definitive but, even today, there is a paucity of in depth shelter data.

         Animal Intake Dogs Licensed                  % Dogs            LA Clinic          # Licensed

         (LA AnServ)                                Sterilized   S/N               Dogs Sterilized

1971         144,530   266,325              -                        0

1972         132,254   253,252             10.9            2,553            27,604

1973         127,554   228,217             26.0            3,065            59,336

1974         118,964   201,275             30.9          10,184               62,194

1975         117,280   245,321             33.1          14,068               81,201

1976         114,363   258,111             35.3          12,665               91,113

1977         104,674   246,525             40.2          11,058               99,103

However, we have estimated (very rough) that somewhere in the region of 100,000 sterilizations were being performed annually in Los Angeles in order to grow the sterilization rate of licensed dogs from 11% to 40% in six years and the bulk of those sterilizations had to have been performed by private practitioners.  There are reports of the sterilization activities of private practices in Los Angeles in the 1974 and 1976 publications of the two national conferences held to address the "overpopulation" problem.   So we are less skeptical of Marsh's claim than the reviewer but are unable to provide any more clarity.

Please check the legend in Figure 6 for some odd characters.

I checked but did not find any problem characters. 

I’m still a bit confused about why cat data appears (separately from dogs) in some places but not others. For example, cats are listed in table 3 but not shown in a graph similar to that in Figure 12.  Please clarify and adjust.

This was an oversight.  The cat data has been deleted.

Figure 12: the title should be sources of dogs, not pets for clarity.   CHANGED

I think that line 463 and line 521 are missing “)”.  Thank you.  We have made the changes.  For the editorial office, the line numbering changes with the edits.  This is a problem for authors trying to respond to reviewer suggestions. 

Is Figure 15 only dogs?  Or all pets? Please clarify in the text, title and figure."  All Pets.  CHANGED.

Round 4

Reviewer 4 Report

The authors have addressed my comments adequately.